# Well-Governed Sustainability and Financial Performance: A New Integrative Approach

**Marian Siminica [1], Mirela Cristea [1] , Mirela Sichigea [1],\*, Gratiela Georgiana Noja [2] and Ion Anghel [3]**

[1]   Department of Finance, Banking and Economic Analysis, Faculty of Economics and Business Administration, University of Craiova, 13 A I Cuza Street, 200585 Craiova, Romania

[2]   Department of Marketing and International Economic Relations, Faculty of Economics and Business Administration, West University of Timisoara, East European Center for Research in Economics and Business, 16 Pestalozzi Street, 300115 Timisoara, Romania

[3]   Department of Financial and Economic Analysis, Faculty of Accounting and Management Information Systems, The Bucharest University of Economic Studies, 6 Piata Romana, 010374 Bucharest, Romania

\*   Correspondence: mirelasichigea@gmail.com; Tel.: +40-732-672-944

**Abstract:** This study investigates the interlinkages between the dimensions of corporate social responsibility (CSR-economic, environmental, social), financial performance (ROA, ROE), and corporate governance (CG), by applying the structural equation modeling technique (SEM). It is based on a sample of 614 large companies from the European Economic Area, covering specific indicators published by the Thomson Reuters database, for the years 2013–2017. The equation models are structured starting from isolated dependencies between variables, up to the global ones (direct, indirect, and total dependencies). The mixed results obtained imply that the nature and heterogenous content of CSR lead to different statistical dependencies for each of the two financial performance indicators. ROA is positively influenced by the economic dimension of CSR, but, the level of this rate does not necessarily contribute to an increase in the involvement of the company in this type of CSR actions. At the same time, ROA is influenced and affects in a negative way the environmental and social dimensions of CSR. In the case of ROE, it is influenced and impacts the economic and social dimensions in a positive way. The environmental dimension of CSR influences ROE positively, but it is negatively affected by this profitability rate. Corporate governance exerts a positive impact on all of the model's variables, both as a direct and indirect factor of influence.

**Keywords:** corporate social responsibility; corporate governance; financial performance; structural equation modeling

## 1. Introduction

The link between corporate social responsibility (CSR) and financial performance (FP), despite the growing body of the research, represents an exciting, unsolved, provocative, and difficult to convey issue for the economic and academic community. The 'on going' stage of all that is related to the conceptualization and applicability of CSR, while permanently modeling the knowledge accumulations of this field of business-society and, at the same time, it poses new questions and issues for scholars. Worldwide, the scientific interest for CSR-FP is being maintained to a high degree through the efforts of various international bodies to harmonize and regulate the practices of information dissemination on the subject of corporate responsibility (Sustainable Development Goals (SDGs), adopted by the United Nations, European Commission, World Business Council for Sustainable Development, The Global Reporting Initiative (GRI), the Carbon Disclosure Project (CDP)). Moreover, the shareholders and/or

stakeholders give a higher increasing significance to information related to CSR topics, enclosing them within the decision-making process of a company [1,2]. The relationship between corporate social responsibility (CSR) and financial performance (FP) has enjoyed a widespread debate among scholars [3]. The review of the empirical research leads to different conclusions, either regarding the existence of a positive link between CSR and FP [4–6], of a negative relationship [7], or even regarding the lack of a link [8] between these variables. The inconsistency is generally caused by the different measures used to cuantify CSR and FP, the heterogenous analysis techniques, the limitations of data (small sample sizes, old data), the features of the sampled companies (dimension, area of activity, affiliation to a developed or emerging market etc.). The mixed and incoherent results themselves have dictated the strengthening and broadening of the approach towards the CSR-FP relationship.

The debates on this subject are far from settled as the stake of the challenge is being kept high by the new research paths that have been discovered. Clarifying this issue implies a deeper exploration, the contestation of all that is doubtful and inconclusive, and retesting causal dependencies. Going further, scholars have begun to direct their attention towards other factors that might influence the causal relationship between CSR-FP, considering the corporate governance (CG), as playing an active role in the companies' decision, to involve themselves in CSR actions [9,10].

On this frame of reference and challenges, the general objective of our research is to build up a new integrative analysis: a mixed model of a global/multidimensional anaylsis of the CG-CSR-FP relationship, by applying the structural equation modeling (SEM) procedure. Similar to the approach of Sahut, Mili, and Teulon [11], we consider that the CSR-FP causal relationship continues to generate numerous scientific debates and that we ought to include in our analysis, as precursory factors bearing influence on CSR actions, the corporate governance and the financial performance indicators. The complexity of our empirical model is also determined by the fact that we have taken into account the multidimensional character of the CSR concept, as well as the heterogenous nature of its dimensions. As a result, CSR will not be quantified using a global index (composed of possibly heterogenous elements), rather we will analyze the individual causality between the three dimensions—economic (ECSCORE), environmental (ENVSCORE), and social (SOCSCORE)—and the financial performance, using the financial indicators ROA and ROE. This causality will be tested in both directions (CSR-FP, as well as FP-CSR). The insertion of the corporate governance in the model (CGSCORE) are impling the analysis both of the direct relationship between it, and the financial performance (CG-FP), as well as the role of mediator for the CSR-FP relationship.

All the variables included in our analysis were extracted from the Thomson Reuters database [12]. Unlike the empirical research of Sahut, Mili, and Teulon [11], we have compiled in our analysis data gathered from a single source (and not from three databases), thus ensuring the comparability and a better quality of the obtained results. In addition, we have ensured a better homogeneity of the data, by including all the sampled companies from the Europe, region that is characterized by a strong harmonization tendency, ensured by the European Union (EU), of the aspects regarding CSR and CG.

This work enriches the knowledge in the field on multiple levels. First of all, it brings a step forward by considering the distinctive dimensions of CSR (economic, environmental, and social), and not mixing them into a single heterogenous index. Secondly, it broadens the analysis of the CSR-FP causal relationship, including the hypotheses of a bidirectional causality between the actions of social responsibility and the financial performance, as well as the influence of CG upon the other variables, thirdly. Fourthly, it ensures an international approach at the level of large companies situated in the European countries. Therefore, the paper enriches the research on this field by an overall assessment of the interlinkages (direct, indirect, and total) among CG-CSR-FP.

The paperis structured as follows: after a detailed introduction regarding the novelty and topicality of the subject, a conceptual framework of the analysis is accomplished, sustained by relevant literature underpinnings. This section comprises, on the one hand, the conceptual presentation of the main variables/concepts used into our research (CSR, CG, and FP), the interlinkages between them studied into other reserachers findings, and our general model, attended by the scientific hypotheses, on the

other hand. Thereafter, we presented the data and panel construction, and the research methodology applied. The results are discussed further, supported by the main findings previously grasped in the literature. Finally, the concluding remarks are enclosed in the Appendix A.

## 2. Conceptual Framework of Analysis

### 2.1. Corporate Social Responsibility (CSR)

Corporate social responsibilty (CSR) is a concept that has been already included within the business practices, as well into academic field for a significant period of time. This concept has been shaped around the 1960 s, when Bowen [13], being considered the founder of CSR, stated in his paper "Social Responsibilities of the Businessman" the necessity for companies to adopt a responsible behavior towards society. The logical foundation for Bowen's [13] convictions was that a society cannot exist and develop without a business scene, and the business cannot exist without society. Followers of the view that CSR has emerged and developed as a result of the interaction between the business sphere and the social one [14], between the internal organizational values and the external influences [15], have been making themselves heard through the years.

The concept of CSR was adopted and quickly spread throughout the literature, the papers of Wood [14,16], Carrol [17,18], and Aupperle [19] being among the research with a significant impact on its development. Even though the meaning, sense, and implications of CSR have been redesigned and redefined by almost every author, the common component of each approach was that firms must integrate, voluntarily, their pursuit with the environment, human rights, and society in their operational activities, as well as in their relationships with other interested parties (client, supplier, local authorities and institutions, community, NGOs, etc.).

Corporate social responsibility acknowledges the importance of growth and development, as well as of profitability, alongside with other objectives of the society [20]. Thus, the concept of CSR represents a strategic, managerial instrument, through which companies seek to maximize their economic performance concomitantly with the social, societal, and environmental aspects. The worldwide issues regarding climate change, world hunger, fossil fuel depletion, have exerted a growing pression on the economies around the world, and have been passed down to the level of their companies. Social, economic, and environmental endeavors oblige companies to integrate systems that consider the common good for society, in general, and for the interested parties, in particular [21]. A socially responsible company distinguishes itself through the manner in which it acknowledges and manages the impact generated upon society and the environment by its activities, taking into account the interests and well-being of its stakeholders, the norms, values, and social expectations, and the interest of the organization itself [22]. In well governed companies, the managers develop a larger number of CSR practices [23], ensure the dissemination and transparency of information of this nature, and involve themselves with obtaining and preserving the confidence of the shareholders and/or stakeholders.

### 2.2. Corporate Governance (CG)

Corporate governance (CG) is also a concept with a rapid growing in the literature. Grounding its origins on the agency theory, CG was initially based on the objective of maximising the value of the firm, by aligning the interests of the shareholders and the managers with the lowest costs [24]. Through this lens, CG represents a set of monitorization, control, and optimization procedures, through which the shareholders ensure the realization of their own interests and objectives. Accepting the possibility that the shareholders and managers can have different objectives, the principles of corporate governance served as a bridge between the managers and shareholders.

In time, the scope and signification of CG has extended to include maximising the interests of all parties involved in the company's activity, shareholders and stakeholders, altghough a significant part of the shareholders of the companies can be important stakeholders of the company. In the latest

studies, CG was redefined as a set of mechanisms that can coordinate and balance the interests of all these parties and can contribute to the growth of benefits for all [25,26]. This evolution of CG has led to confusion and conflicts of the interest regarding the concept of CSR, however, there are also opinions that claim that the connection between CG and CSR itself is responsible for the development of the principles of CG [21,27]. In the same manner, CG, with its predominant elements (management and property structure), represents a key factor in the decision-making process, concerning specifically those decisions that regard actions and policies in the field of CSR.

## 2.3. Financial Performance (FP)

Financial performance reflects the capacity of a company to achieve its profitability targets, being the standard landmark used for assessment of any economic activity. Financial performance indicators (as accounting or stock market dimensions) represent significant tools for the stakeholders' decisions process. In the complexity of the contemporary business environment, the stakeholders are forced to pursue much more in depth than the profit level of their company. They need to understand the impact that the company is having on the environment and society, and to perceive how conscious and responsible the company is in acknowledging and optimizing this impact and the related risks. Size and financial power are no longer being regarded themselves as guarantees of the medium- and long-term success of a company. These concepts ought to be integrated into a much larger picture.

With the growing of global interest for environmental and social issues, companies have reacted and prioritized the environmental and social factors as an integrated part of their corporate strategy. The result has led to a symbiosis between the goal of profit maximization and that of corporate responsibility. In other words, the responsible behavior of companies endorses the achievement of their profitability objective, while their financial power enables the accomplishment of CSR practices.

In order to look into the relationship between financial performance and corporate social responsibility, the return on assets (ROA) and the return on equity (ROE) are the indicators most commonly used for companies' performance measure [28]. Both indicators are determined according to the accounting data and they reflect the relative rentability generated by the employ of the company's assets (ROA), and by the avail of the company's equity (ROE). These accounting and financial indicators (ROA, ROE) are used also to explore the relationship—both direct and indirect—between financial performance and corporate governance [9–11,23]. Numerous scholars have identified a statistically significant positive relationship between FP and CSR due to the significance given by companies to the natural resources in their economic activity, and to the socially responsible activities under the guidance of a successful management [22].

## 2.4. Applied Model and Interlinkages among CSR-FP-CG

### 2.4.1. General Conceptual Model

As we have previously underlined, the main objective of this study is to analyze the causal links that arise between the considered variables: corporate social responsibility (CSR), financial performance (FP), and corporate governance (CG), at the level of large European companies. The conceptualization of these relationships is explained by the model enclosed in Figure 1.

The central axis of the conceptual model represents the causal linkages between CSR-FP. These linkages are approached in both influence directions (CSR-FP, as well as FP-CSR), seeking answers to the following dilemma of economic practice: does the solid financial performance of a company represent a decisive factor for the intensity of CSR actions? Do these actions (CSR) contribute with the same intensity to the growth of the financial performance of the company?

The conceptual model is completed by the influence that CG, as an expression of the key factors in the decision-making process of a company, can exert both directly upon FP, as well as mediator for the CSR-FP relationship. Another feature of the theoretical model consists in including the three individual dimensions of CSR (economic, environmental, and social), rather than a general score of

CSR. CSR decomposition into several dimensions contributes for a better understanding of the concept and a more accurate representation of its multidimensional nature. In order to quantify the financial performance of the companies, we use the standard indicators, ROA and ROE. The scientific basis of the causal links proposed by the research is being presented in the following paragraphs.

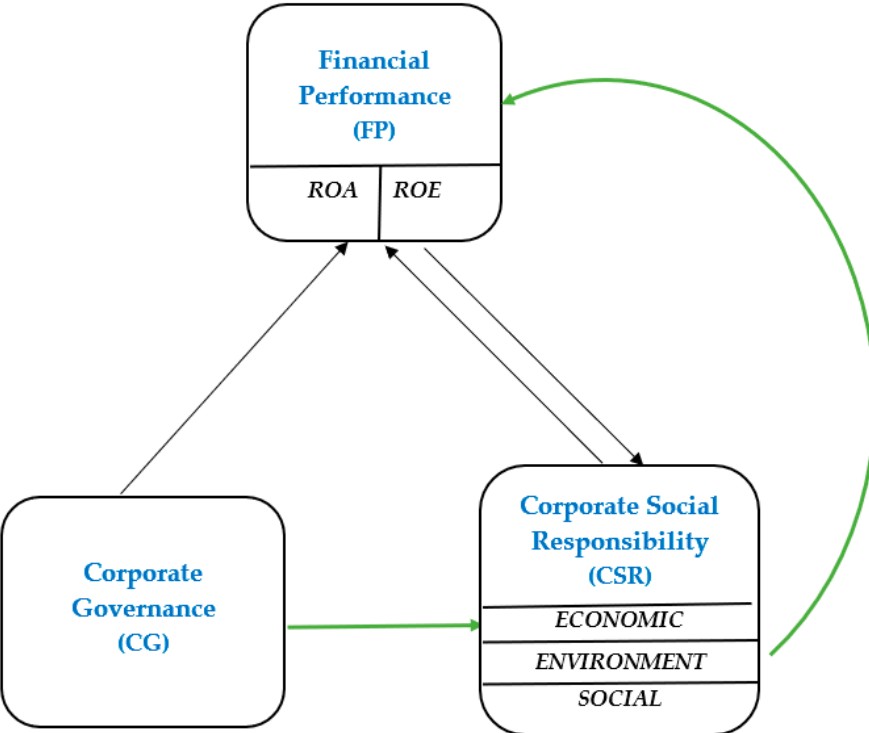

**Figure 1.** Conceptual model of CSR-FP-CG. Source: own contribution.

2.4.2. Bidirectional Linkages between CSR-FP

The existence of a causal dependency between corporate social responsibility and the financial performance of a company is supported by the vast evidence from the literature [2–8,11,28,29]. However, the direction and intensity of this connection are still causing controversy and dispute among scholars. Most of the evidence points to a positive link between CSR-FP [3–6], claiming that the involvement in CSR actions contributes to a growth of the company's financial performance. Most of these empirical studies assumed a direct link, in a single direction: CSR-FP, ignoring certain valences of economic practice. The theoretical models proposed by these studies only analyzed the manner in which CSR leads to FP, without taking into account the reality that large companies, with a solid financial performance, are typical leaders when it comes to undertaking and reporting CSR actions, their behavior creating vast trends that are then adopted by the other companies [30].

The hypothesis of the interdependences between the social and financial variables has been addressed by several researches. Rodriguez-Fernandez [21] has analyzed this interdependence at the level of the Spanish companies listed at the Madrid Stock Exchange. The empirical results, using the multivariate regression models, have demonstrated a positive bidirectional CSR-FP relationship. In other words, the intensity of CSR policies contributes to the growth of the financial performance, just as the increasing of financial performance leads to a higher corporate social responsibility [21]. The existence of a bi-dimensional CSR–FP relationship has been tested at the level of Fortune 100's Most Admired Companies. The research, based on the panel analysis technique (Panel VAR), has proven a strong positive connection between FP and CSR (a higher financial performance leads to superior CSR involvement), and a strong negative link between CSR and FP (superior CSR actions do not lead to a growth in financial performance, but, on the contrary, to a decrease in profit) [31].

Both studies [21,31] utilized global indexes in order to quantify CSR and the traditional ROA, ROE measurements in establishing financial performance.

Two other representative studies [11,28] have confirmed the existence of the connection between CSR-FP, but with different trajectories (positive and negative). Hirigoyen and Poulain Rheim [28] have concluded that, based on the linear regression analysis and on the Granger causality test, the components of CSR-FP influence negatively each other. They remarked the fact that the concept of CSR promotes a multidimensional and comprehensive idea, with scholars having to pay great attention to the different nature of the dimensions that make up CSR. The research has targeted the investigation through an empiric lens of the bidirectional relationship between the dimensions of CSR (economic, environmental, social) and financial performance (return on assets (ROA), return on equity (ROE)). The study was carried out at the level of large companies from three regions, the United States of America (USA), Europe, and Asia-Pacific. Meanwhile, the French authors Sahut, Mili, and Teulon [11], have demonstrated—following an empirical study carried out at the level of large companies from Europe and the USA—that CSR and FP positively influenced each other. These studies [11,28] are representative for our research, not only because they provide a global analysis (covering extended regions), but especially due to the fact that they use the individual dimensions of CSR in its quantification, rather than a global index. Both studies have based their conceptual models on the opinion that CSR is a multi-dimensional concept, with heterogenous components that ought to be analyzed independently [11,28]. In addition, the lack of a consensus regarding the CSR-FP link suggests that only certain particular combinations of environmental, social, and societal practices influence the financial performance of a firm [11]. Concerning financial performance, the ROA and ROE indicators have been utilised by both studies.

Using a unique index of CSR that mixes the different heterogenous dimensions (economic, social, environmental, employees, human rights) may not be the best solution for the analysis of the dependent relationship between CSR and FP. At the same time, the following question ought to be asked: does CSR alone influence FP or, for a good implementation and involvement in CSR actions, must companies possess a prior solid financial performance? It is a known fact that the CSR trajectory is defined by large companies, multinationals with well-established financial status.

Thus, in agreement with the main findings of the previously empirical research, by extension, with the paradigm of interdependence within economic practice, namely that the business-society components engage in sustained mutual interactions, we expect that CSR and FP are influencing each other positively. Subsequently, we propose the following hypotheses:

**Hypothesis 1A (H1A).** *Performance growth experienced by large companies leads to superior corporate responsibility;*

**Hypothesis 1B (H1B).** *Performance growth experienced by large companies does not lead to superior corporate responsibily, but rather to its decreasing;*

**Hypothesis 2A (H2A).** *Intensive corporate social responsibility leads to a growth in the financial performance of large companies;*

**Hypothesis 2B (H2B).** *Intensive corporate social responsibility does not lead to a growth in the financial performance of large companies.*

### 2.4.3. Interlinkages among CG-CSR-FP and CG-FP

In the context of empirical research concerning CSR, it is important to take into consideration the key decision-making groups, such as the executive directors and the property structure, respectively the components of corporate governance (CG). These key decision-making groups have the power to control the operational activities of the company and the policies adopted by it, including those

pertaining to CSR, for a better understanding of both CSR, as well as FP [32]. In fact, the mixed and ambiguous results concerning the CSR-FP connection have led scholars to widen the the hypothesis range and to also take into consideration the indirect, mediating factors of the causal relationship between the two variables. Corporate governance is considered the most important factor of this type [33–35].

CG and CSR have been studied separately in the literature, but the relationship between them generated beneficial synergies [21] for the company's performance. The number of authors who have approached this type of mediating relationship is ever growing. By analysing the data from the public companies listed by Pakistan's sector of production, the authors [9] have observed that CG exerted a powerful mediating effect, consolidating the link between CSR and FP. Another study that advanced the intermediary effect of CG on the CSR-FP link [10], has generated—based on a sample of 155 firms from a developing country—empirical results which showed the influence of corporate governance (CG) and corporate image (CI) on this connection. Corporate governance exerted an influence on FP that was not just indirect, via synergy with CSR; but also direct, as a result of the decisional power that it can exert on a company's operational strategy [11,21,23].

Relying initially on the goal of maximizing the value of the firm (in accordance with the agency theory), therefore on the realization of the shareholders' interests [24], corporate governance has evolved towards encompassing in its sphere of activity the interests of the stakeholders as well (altghough a significant part of the shareholders of the companies can be important stakeholders of the company), creating a nexus between CG and CSR. The corporate social responsibility (CSR) and the corporate governance (CG) are two different concepts, both in terms of definition as well as origin, but their evolution, overlapping at times, has implied interactions between their concepts. In a review of the comparative literature of shareholders and stakeholders' theories, Money and Schepers [27] have proved that the scope of the corporate governance has been expanded in time as a result of the ever-growing influence and importance of CSR. However, while CG refers to a system of mechanisms, processes, practices, and rules, through which corporations are controlled, monitored, and led [25], with the purpose of maximising the value of the firm by achieving the objectives of the shareholders and stakeholders, the concept of CSR deals with the manner in which the companies engage with social and environmental issues [36]. Both individually, but especially together, these concepts unite the interests of the shareholders and stakeholders, leading to the creation of the competitive advantages and, implicitly, to superior financial performances for the companies.

CSR actions seem to rely on the financial resources, in order to satisfy the interests of the involved parties, as well as on the power dynamic between these parties [37]. An integral component of the concept of sustainable development, corporate responsibily, continues to mature as "a management process involving the efforts that a corporation takes to behave responsibly toward society" [25] (p. 215). Organizational culture and the practices of corporate governance on a company level play an important role in this context. In fact, the power to engage in CSR actions is dependent on the governing bodies of the company [11]. Therefore, it is vital for the analysis of the relationship between CSR-FP to also encompass aspects pertaining to corporate governance (executive directors, property structure) including here the authority that controls the operational and social activity of the companies, the one that regards the company's performance in its full context (financial performance and social performance).

Consequently, we propose a second set of hypotheses:

**Hypothesis 3A (H3A).** *Corporate governance mediates the link between corporate social responsibility and the financial performance of large companies;*

**Hypothesis 3B (H3B).** *Corporate governance does not mediate the link between corporate social responsibility and the financial performance of large companies;*

**Hypothesis 4A (H4A).** *Good corporate governance leads to a growth in the financial performance of large companies;*

**Hypothesis 4B (H4B).** *Good corporate governance does not lead to a growth in the financial performance of large companies.*

## 3. Materials and Methods

### 3.1. Data and Sample Construction

The companies and the variables included in the empirical research were obtained by consulting the Thomson Reuters database [12]. Since it contains the largest sample of companies from all the regions of the globe, having the oldest (longest) chain of historical data, Thomson Reuters provides to its users, with efficient managerial instruments, starting from standard financial indicators, to those specific of financial and investment markets, to mergers and acquisitions, up to information on risk and the ESG (economic social governance). From the variety of products and instruments provided by the Thomson Reuters database, we took interest in and extracted only those indicators that are specific to our conceptual model: (i) ROA and ROE, as measurements of financial performance; (ii) economic, environmental, and social scores, as specific elements of corporate social responsibility; (iii) and the corporate governance score.

Out of more than 7000 companies from around the globe, about which the Thomson Reuters database lists information concerning the environmental, social, and economic indicators, as well as the corporate governance indicators, we have selected only companies from the European Economic Area (EU Member States, Iceland, Liechtenstein, and Norway), obtaining an initial sample of 1098 companies, with indexed values specific to the timespan of 2002–2017.

Out of the 1098 total companies initially selected, we were included for the empirical analysis only the ones that contained complete data sets for all the indicators of the conceptual model. Moreover, financial and credit companies were eliminated as a consequence of the specificity of their financial performance indicators. Thus, the final statistical sample size comprised a total number of 614 companies, with complete values for the all six variables of the model, relevant for the time frame 2013–2017. The structure of the sample used in our research is presented in Table 1.

**Table 1.** Sample.

| Country | No. of Companies | Country | No. of Companies |
|---|---|---|---|
| Austria | 9 | Ireland | 23 |
| Belgium | 20 | Italy | 24 |
| Cyprus | 1 | Luxembourg | 7 |
| Czech Republic | 2 | Netherlands | 32 |
| Denmark | 17 | Poland | 13 |
| Finland | 23 | Portugal | 6 |
| France | 73 | Spain | 31 |
| Germany | 63 | Sweden | 36 |
| Greece | 8 | United Kingdom (UK) | 223 |
| Hungary | 3 | | |

Source: own investigation.

The 614 companies that make up the surveyed sample belong to a total of 19 countries of the EEA. The UK is the country with the highest number of companies (223), followed by France (73), and Germany (63). On the opposite side, we have Cyprus with only one company, and the Czech Republic with two companies.

The decision to include in our analysis only companies from the European Economic Area (EEA) was based on the need to ensure the homogeneity and comparability of the data included in our

research, and to offer findings for a future kindred/comparative research, after applying the European Union (EU) Directive 2014/95/EU [38], outset by 2017 (after our timespan analysis). Presently, the EU is an international leader when it comes to regulations concerning the publishing of non-financial information and diversity. Thus, starting with 2017, the implementention of Directive 2014/95/EU represented a significant step made by the EU towards both the harmonization of CSR across its Member States, and "to legitimise non-financial reporting (NFR) that encompasses two grand theories: improve the comparability of information and enhance corporate accountability" [39] (p. 598). Although this step is addressing to public companies with over 500 employees, the directive's implementing at the national levels has determined, on the one hand, the increasing of the companies' number that publishes this type of information, and "to confirm the role of regulation in improving the quality of disclosure of non-financial information" [40] (p. 1). On the other hand, some scholars [41] (p. 11) have revealed that this directive, which is based on the agency theory, does not reshape the managers' behavior, "nor to act in the interest of all stakeholders as the stewards of an organisation's resources". In this vein, changing the managers' behavior becomes more significant than any other theories, "agency, legitimacy and stakeholder theories" [41] (p. 11).

*3.2. Variables*

The indicators used for the empirical research correspond to the three categories of variables described in the conceptual model, namely: corporate social responsibility, financial performance, and corporate governance. The elements of each category of variables, as well as the adequate way to express these elements, are detailed in Table 2.

**Table 2.** Variables.

| Category | Name | Symbol | Type |
|---|---|---|---|
| Corporate Social Responsibility (CSR) | Economic dimension | ECNSCORE | Score (Scale: 0–100) |
| | Environmental dimension | ENVSCORE | Score (Scale: 0–100) |
| | Social dimension | SOCSCORE | Score (Scale: 0–100) |
| Corporate governance (CG) | Corporate governance | CGVSCORE | Score (Scale: 0–100) |
| Financial performance (FP) | Return on assets | ROA | Net result/Total assets |
| | Return on equity | ROE | Net result/Equity |

Source: own contribution.

The relevant variables, CSR and CG, are expressed in the form of scores, on a scale from 0 to 100. The economic, social, and environmental dimensions and the corporate governance are calculated in accordance with Thomson Reuters' methodology, by combining more than 250 key performance indicators amongst 18 groups of indicators that correspond to the four dimensions. In line with the methodology and with the available information provided by the Thomson Reuters database, the informational content is summarised as follows [12].

Economic (ECNSCORE) measures the global performance of a company, the state of its financial health. It includes information such as client loyalty, the performance and loyalty of its shareholders and expresses the capability of a company to generate long term value, sustainable growth, and high profitability rate on its investment, by efficiently exploiting the available resources.

Environmental (ENVSCORE) characterizes the ability of a company to reduce environmental risks, as a consequence of adequate environmental practices that are oriented towards reducing emissions, resources, and towards an increase in innovative products. A high regard for environmental opportunities leads to growth in long term value.

Social (SOCSCORE) measures the capability of a company to generate good social reputation, trust, and loyalty on the part of the employees, clients, suppliers, and community as a whole. By applying the best management practices when it comes to the safety and health of the employees, their training, human rights, the community, and the responsibility of the product, the social pillar establishes the capability of the company to generate long term value, especially significant for the shareholders.

Corporate governance (CGVSCORE) characterizes the ability of a company to implement the best systems and governing processes, through which to ensure that the council members and its directors act in the long-term interest of their shareholders.

### 3.3. Empirical Approach

In order to highlight the overall interlinkages (direct, indirect, and total) between the variables of corporate social responsibility (CSR), financial performance (FP), and corporate governance (CG), we have employed the method of structural equation modeling (SEM) [9].

Structural equation modeling (SEM) is a diverse technique of statistical research, which combines factor analysis with multiple regression analysis. The SEM technique is used to analyze complex systems of variables, with numerous intercorrelated dependencies. This encompasses both direct links between variables, as well as indirect, and total ones.

The SEM analysis is applied to the conceptual model (Figure 1), built on specialized theory and the results of previous empirical research. The specific tests will indicate in which way the hypotheses correspond to the model, based on the analyzed data, validating or invalidating them [42].

## 4. Results and Discussions

### 4.1. Panel Processing

Analysing the results of the descriptive statistics presented in Table 3, the scores published by the Thomson Reuters database for the four dimensions specific to the CSR and CG of all the companies of the 19 European states surveyed, a high average performance can be noticed. Averages of around 70 points have been registered for all CSR dimensions, the highest level being represented by the social pillar (SOCSCORE = 74.97 points), followed by the environmental pillar (ENVSCORE = 72.40), and the economic pillar (ECNSCORE = 69.75).

**Table 3.** Descriptive statistics.

| Variables | N | Mean | Standard Deviation (sd) | Min | Max |
|---|---|---|---|---|---|
| ECNSCORE | 3070 | 69.75725 | 24.49642 | 1.26 | 98.52 |
| ENVSCORE | 3070 | 72.40101 | 25.46606 | 8.38 | 95.5 |
| SOCSCORE | 3070 | 74.97696 | 23.01117 | 3.77 | 97.49 |
| CGVSCORE | 3070 | 65.86599 | 24.76057 | 2.12 | 98 |
| ROA | 3070 | 4.85738 | 7.818485 | −55.944 | 67.931 |
| ROE | 3070 | 10.67942 | 30.91214 | −656.566 | 274.653 |

Source: own research.

Concerning the CG, European companies have reached, on average, a score of 65.86 points (Table 3). Overall, these values demonstrate a higher interest of the analyzed companies for the three dimensions of CSR, compared to the aspects related to corporate governance.

Before testing the existing dependencies among CSR-FP-CG, we have stationarized the variables of the conceptual model through logarithm, due to the fact that the unit root test results showed that some series were non-stationary.

Using the logarithmized data of all 614 major companies from countries belonging to the EEA, we have built several models of structural equations in order to test:

- the bidirectional relationship between the dimensions of CSR (economic, environmental, social) and the financial performance (ROA, ROE);
- the influence exerted by corporate governance (CG) on financial performance (ROA, ROE) both indirectly, as mediating factor for the CSR-FP relationship, and as a direct factor of influence;



- the total, direct, and indirect dependencies among the three categories of variables, namely CSR (economic, environmental, social), financial performance (ROA, ROE), and corporate governance (CG).

The structural equation models were configured using the Stata software.

*4.2. Bidirectional Relationship CSR-FP*

The dependencies tested thorugh this model correspond to the first set of proposed scientific hypotheses, and are based on the arguments of scholars, such as "CSR can be both a consequence as well as a determining factor of financial performance" [43] (p. 186). In the case of both subsets of hypotheses H1 and H2, we have considered the existence of a connection between variables as the null hypothesis.

Figure 2a,b shows the estimated coefficients between the CSR variables (economic—ENCSCORE; social—SOCSCOR; environment—ENVSCORE) and the financial performance expressed, progressively, through ROA and then ROE, indicating the power and trajectory of this dependency, thus:

- ROA is positively bi-correlated with the economic dimension of CSR (coefficient = 0.0032), however the link is not significant from a statistical point of view. At the same time, ROA is negatively bi-correlated with the social dimension of CSR (estimated coefficient = −0.045, statistically representative for the 0.1% threshold), but also with the environmental dimension (−0.057 coefficient, statistically representative at the same threshold of 0.1%). These dependencies disprove the first set of hypotheses, regarding a significantly positive bidirectional link among the variables (H1 and H2). In other words, the alternative hypothesis, the one concerning a negative bidirectional link, or an inversely proportional one, an evolution in opposite directions of the analyzed variables is being confirmed. Through an economic lens, the trajectory of these dependencies suggests that the involvement of companies in social and environmental actions leads to a decrease in profitability, as a result of the costs associated with this endeavor;
- ROE is positively bi-correlated with the economic dimension of CSR (the estimated coefficient of 0.041 being statistically representative for the 0.1% threshold). The estimated coefficient of the linkage between ROE and the social dimension (SOCSCORE) is a positive one (0.015), but also insignificant from a statistical point of view. Likewise, insignificant is the estimated coefficient corresponding to the bidirectional dependence between ROE and the environmental dimension of CSR. We must observe that this negative coefficient (−0.0055), implies that the trajectory of this relationship is inversely proportional. In the case of using ROE as the measurement of financial performance, the first set of hypotheses is partially validated, a significant, positive, bidirectional link is only validated in the case of the relationship between the economic dimension of CSR (ECNSCOR) and ROE. From an economic standpoint, this connection implies that a more adequate satisfaction of the clients and associates of a company (as components of the ECNSCORE) leads to an increase in ROE and vice versa, a company with superior profitability is much more involved in the satisfaction of its clients' and shareholders' needs.

Summarizing the results of these equation models, we can affirm that ROE is influenced and influences positively the economic dimension of CSR (ROE-ECNSCORE: positive, bidirectional, and statistically significant correlations). In the case of ROA, the significant correlations are negative and are established with the social and environmental dimensions of CSR. All other combinations of dependencies are not statistically significant. It is important to note that a significant, negative impact, in the case of the CSR-FP relationship, was also observed throughout different studies [28,31]. It is however necessary to state that these are only partial equation models, the scientific hypotheses need to be tested in the context of a global model that encompasses all dependencies among variables, both direct and indirect.

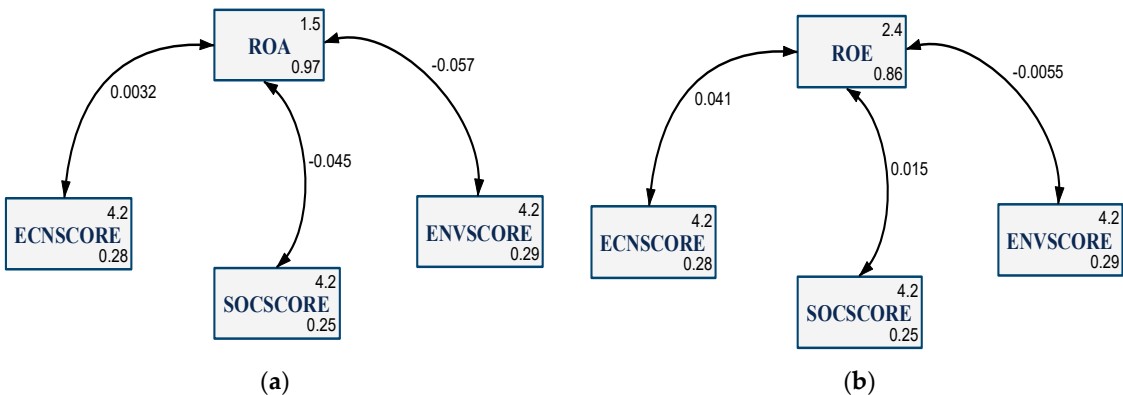

(**a**)                                                                                   (**b**)

**Figure 2.** Bidirectional influence between CSR and ROA (**a**), respectively CSR and ROE (**b**). Source: own research in Stata.

Therefore, *the first set of hypotheses—H1 and H2—is partially validated*.

### 4.3. CG Influence on the CSR-FP Relationship

In the relationship between the financial performance of a company and its CSR policy, the decisional factors represent the binding or mediating factor. The management team, the company's board, the shareholder structure, as well as the entire organization of the decision-making process play a vital role in improving the relationship between CSR and FP [32]. All of these are elements of corporate governance, CG being the most significant mediating factor of the CSR-FP link.

Having deemed as null hypotheses the statement "corporate governance mediates the link between corporate social responsibility and the financial performance of large companies" ($H_{3A}$), and, at the same time, "good corporate governance leads to a growth in the financial performance of large companies" ($H_{4A}$), structural equation models have been generated for each combination of financial performance indicators, ROA and ROE, and each dimension of CSR (economic, social, and environmental).

The results show that corporate governance exerts an indirect, positive influence over all three categories of connections between CSR-FP, as follows:

- indirect positive influence (through a statistically significant coefficient of 0.23) of CG on the bidirectional ROA-ECNSCORE (a coefficient of 28) dependence (Figure 3a). An increase in the value of the coefficient associated with this link from 0.0032 (in the case of the isolated dependence between the variables) (Figure 2a) to 28 (in the case of a link mediated by CG) highlights the major role that CG has in consolidating the link between the two categories of variables;
- the indirect positive influence, suggested by the 0.22 coefficient, that CG has over the ROA-ENVSCORE dependence (a coefficient of −1.9 reiterates the reverse bidirectional link between these variables) (Figure 3b). Moreover, we observe the growth in size of this coefficient and, implicitly, in the power of the link between the variables;
- the indirect positive influence, with an estimated coefficient of 0.28, that CG exerts over the negative, bidirectional dependence between ROA-SOCSCORE (a coefficient of −4.7 consolidates this dependence) (Figure 3c).

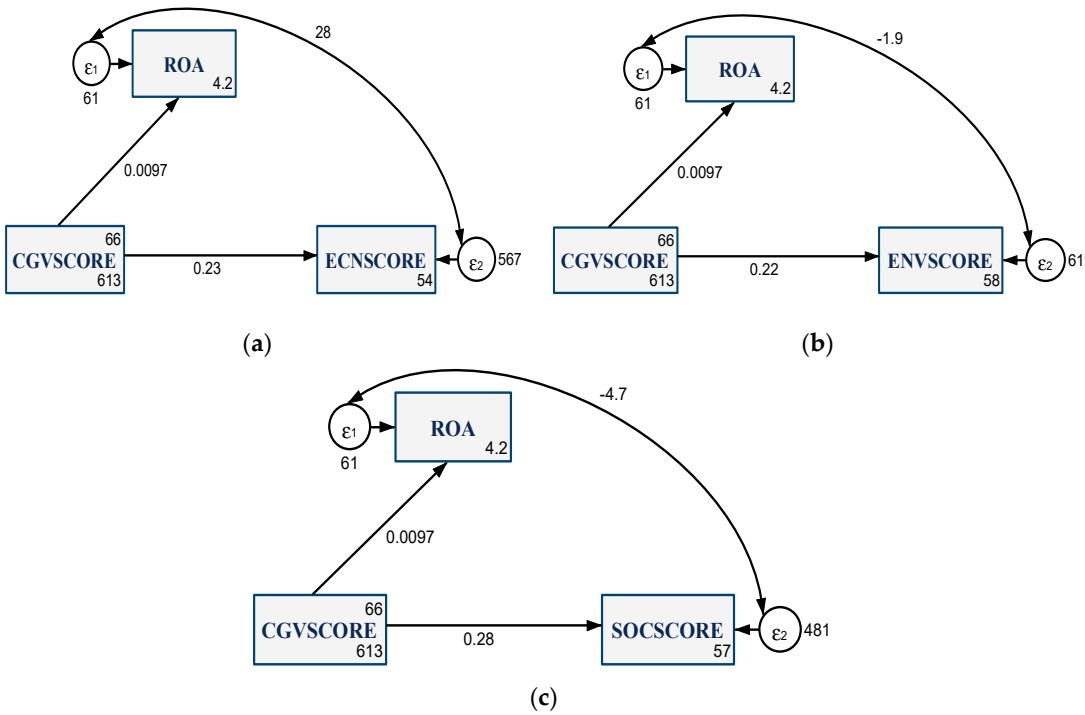

**Figure 3.** SEM results for CG-ROA, combined with CSR dimensions: economic (**a**), environmental (**b**) and social (**c**). Source: own research in Stata

At the same time, corporate governance also exerts a direct, positive influence (with a coefficient of 0.0097) over the size of a company's financial performance (expressed through ROA) (Figure 3a–c).

Likewise, in the case of expressing the financial performance of the companies through ROE (Figure 4), we observe that CG exerts an indirect, positive influence over the three series of bidirectional dependencies, as follows:

- the indirect positive influence, expressed by a coefficient of 0.23, over the ROE-ECNSCORE (the estimated coefficient registering the same growth tendency as in the case of ROA, up to a value of 94) confirms the role of mediator that CG also has in relation to this link (Figure 4a);
- the indirect and positive influence, determined by a coefficient of 0.22 concerning the ROE-ENVSCORE dependence (Figure 4b). We must take into account that the bidirectional link between these variables is a positive one (the estimated coefficient being 4), unlike in the analysis of the variables of the ROE-ENVSCORE dependence by themselves, where the results showed a negative link (with a coefficient of −0.0055) (Figure 2b). The economic implications of this connection point out the role that CG can have in improving the relationship between financial performance and the environmental policy of a company. Adopting adequate environmental strategies, with controlled costs, can lead to the growth of ROE;
- the direct positive influence (a coefficient of 0.28) of CG over the bidirectional ROE-SOCSCORE dependence (the coefficient of 3.6 confirming this link) (Figure 4c).

Corporate governance also influences ROE directly, through the 0.0929 coefficient (Figure 4a–c).

Therefore, *the second set of hypotheses—H3 and H4—is fully validated*, both when utilizing ROA as a measurement of FP, as well as when utilising ROE. By proving that corporate governance mediates the CSR-FP link in a significant way, the research contributes, along with other relevant studies, to the evolution of the models that have assessed this dependence.

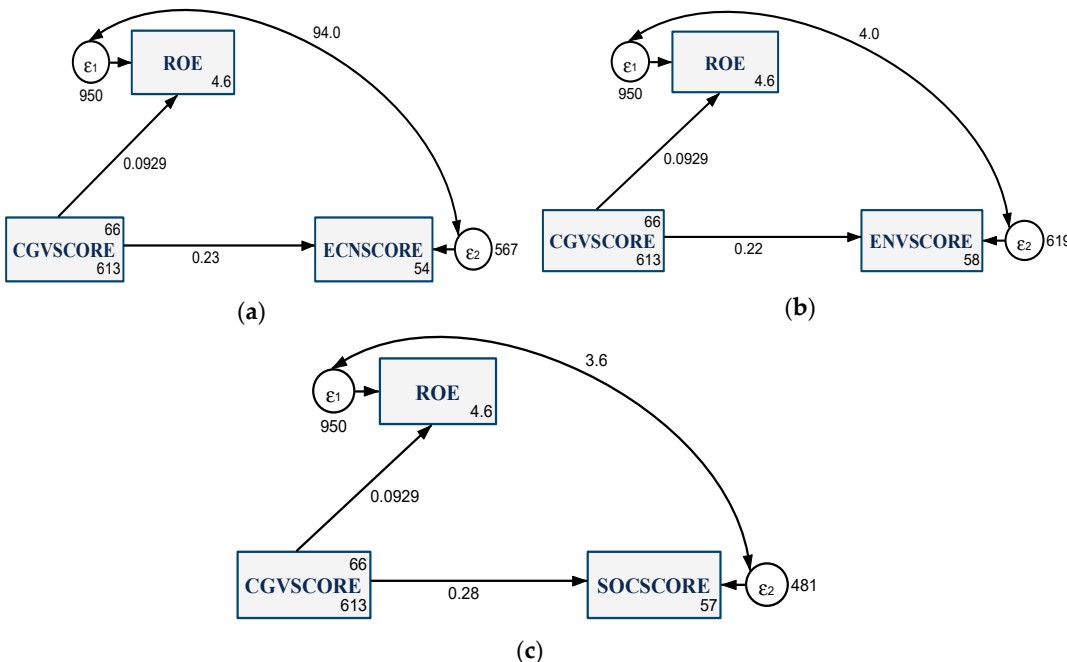

**Figure 4.** SEM results for CG-ROE, combined with CSR dimensions: economic (**a**), environmental (**b**), and social (**c**). Source: own research in Stata.

*4.4. Global Model of Equations*

In addition to testing the two sets of hypotheses that concern the partial equation modeling between variables, we have also generated a global equation model that includes all dependencies (direct, indirect, and total) between the variables of the conceptual model. The SEM technique is recommended in the analysis of models containing complex equations, with multiple interdependencies.

The results of the global model of analysis of the dependencies between the variables of CG-CSR-FP, using ROA and ROE as dependent variables, reflects the following links (Figure 5):

- ENCSCORE-ROA shows a positive, unidirectional dependence (the estimated coefficient of 0.17 is very significant from a statistical point of view, at the 0.1% threshold). The economic dimension of CSR significantly influences ROA, in the direction of its growth, proving that a good policy concerning the satisfaction of clients and shareholders leads to a superior profitability of the company's assets. The influence of CG (with a statistically significant coefficient of 0.25) is included in the dependence, as an indirect factor or mediator of the mechanism, through which management strategies determine the increase of ROA, as a consequence of adequate economic policies;

- ENVSCORE-ROA reflects a negative, unidirectional dependence (the estimated coefficient being −0.2, very statistically significant at the 0.1% threshold). This influence implies that the involvement and implementation into environmental actions leads to a decrease in the profitability of a company's assets. Likewise, we also observe the indirect, positive influence of CG (with an estimated coefficient of 0.29, which is very statistically significant);

- SOCSCORE-ROA shows a negative, unidirectional dependence (through the estimated coefficient of −0.18, significant at the 0.1% threshold). Thus, the social dimension of CSR significantly contributes to a decrease in ROA, through any increased involvement in social and societal policies (employees, clients, suppliers, community, etc.). An important role in this dependence is attributed to corporate governance, as a factor of influence of the second degree, which mediates the adoption of strategies that concern social policy and their integration in the global business strategy of a company;

- CG-ROA reflects a direct positive dependence (with an estimated coefficient of 0.12 being very statistically significant), thus corporate governance influences ROA in a favorable way;

- ECNSCORE-ROE, positive, unidirectional dependence (the estimated coefficient of 0.18 is very statistically significant at the 0.1% threshold), through which the profitability of a company's equity is positively influenced by the economic dimension of CSR. The growth in ROE under the influence of the ECNSCORE is also attributed to the indirect influence of CG, namely the way in which policies regarding the satisfaction of the clients and shareholders is implemented at the level of the company;

- ENVSCORE-ROE, negative, unidirectional dependence (the estimated coefficient being −0.17 and statistically significant at the 0.1% threshold), the environmental dimension of CSR exerting a negative influence on ROE, leading to a decrease in its size, as a consequence of the increased involvement in this kind of policies. The mediating influence of CG is statistically significant as well as positive in the case of this dependence, the decision to reduce pollution and implement innovative, environmentally-friendly product strategies belonging to the governing body of the company;

- SOCSCORE-ROE, positive, unidirectional dependence (with an estimated coefficient of 0.021 being statistically significant), the social dimension exerts a positive influence over ROE, generating its growth, while the influence registered over ROA has proven negative. This dependence suggests that the involvement of a company in social and societal actions is positively appreciated by the present and potential shareholders of the company, leading to an increase in the profitability of a company's assets. Elements pertaining to CG confirm their mediating role for this dependence (the estimated coefficient being 0.34 and statistically significant);

- CG-ROE, direct, positive dependence (the estimated coefficient of 0.16 being statistically significant), corporate governance influencing ROE in the direction of its growth.

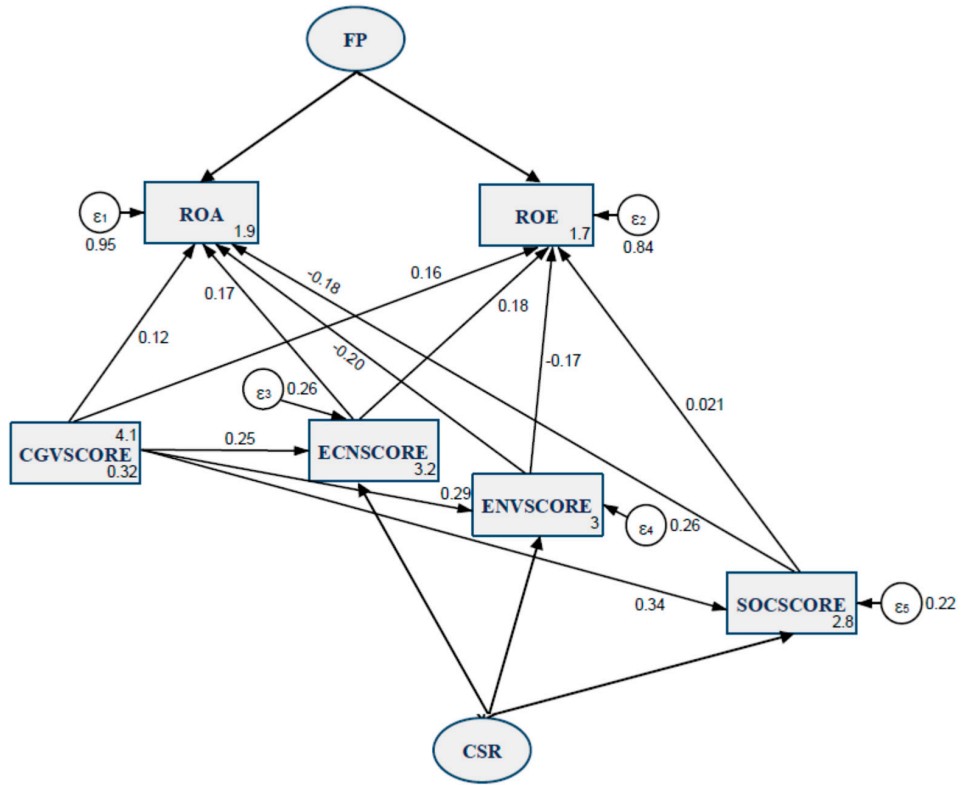

**Figure 5.** SEM results for CG-CSR-FP. Source: own research in Stata.

The results of the global analysis model for the dependencies between the variables of the CG-FP-CSR link—when using ECNSCORE, ENVSCORE, and SOCSCORE as dependent variables—reflect the following links (Figure 6):

- ROA-ENCSCORE, negative, unidirectional dependence (the estimated coefficient of −0.11 being statistically significant at the 0.1% threshold), ROA influences the economic dimension of CSR in a negative way, showing that a greater profitability of a company's assets does not represent a guarantee of the involvement of thecompany in economic policies. The dependence is also being mediated by the indirect influence of CG on the economic dimension of CSR (with a coefficient of 0.045, yet statistically insignificant);
- ROA-ENVSCORE, negative, unidirectional dependence (the estimated coefficient being −0.17 and statistically significant at the 0.1% threshold). The economic profitability of assets significantly influences the environmental dimension of CSR, leading it to a decrease. When observing this dependency, we also note a positive, indirect influence on the part of CG;
- ROA-SOCSCORE, the negative, unidirectional dependence (with an estimated coefficient of −0.19, representative at the 0.1% threshold). The social dimension of CSR is also negatively influenced by ROA, under the indirect influence of corporate governance;
- In the case of CG-ROA, we confirm a direct, positive dependence, with an estimated coefficient of 0.045, although not very significant from a statistical point of view;
- ROE-ECNSCORE, positive, unilateral dependence (the estimated coefficient being 0.13, very statistically significant at the 0.1% threshold), the profitability of a company's assets positively influencing the economic dimension of CSR and implying that an increase in ROE generates better involvement on the part of the company in CSR actions. This dependence is also indirectly amplified by CG (with a coefficient of 0.17 being statistically significant);
- ROE-ENVSCORE, positive, unilateral dependence (the estimated coefficient being 0.13 and statistically representative at the 0.1% threshold). The environmental dimension of CSR is positively influenced by ROE, showing that higher profitabily of assets leads to an increase in the involvement in actions concerning environmental protection. Likewise, we notice the mediating influence of CG over ROE;
- ROE-SOCSCORE, positive, unidirectional dependence (with an estimated coefficient of 0.18, statistically representative). ROE exerts a positive influence over the social dimension of CSR, generating its growth. Likewise, elements pertaining to CG confirm their mediating role regarding this dependence;
- CG-ROE, positive, direct dependence (estimated coefficient of 0.17 being statistically representative), corporate governance influencing ROE in the direction of its growth;
- In both situations (ROA and ROE), we notice the favorable, direct influence of CG over ECNSCORE (the 0.24 coefficient is very statistically significant), ENVSCORE (statistically significant coefficient of 0.28), and also SOCSCORE (estimated coefficient of 0.32 being statistically significant).

Summarizing these results, we have oserved that an upper economic dimension of CSR leads to a growth of ROA, but expanding ROA does not necessarily guarantee a stronger involvement of the company in actions that satisfy the interests of the clients and shareholders. The environmental and social dimensions of CSR influence and are unfavorable influenced by ROA. The companies' involvement into social, societal, and environmental actions contributes to a decrease of the profitability of assets, and vice versa, the level of ROA is not a factor for the increase of actions concerning security and health in the workplace, community, human rights, pollution reduction, and developing innovative products.

In the case of ROE, the links that we have established with each of the CSR dimensions, have led to slightly different conclusions and economic implications. ROE is influenced and positively influences two of the CSR dimensions, namely the economic and social ones. Profitability of assets grows under the impact of the benefits generated by the social and societal policies, and those regarding clients' satisfaction. ROE leads, also, to the enhancement of the companies' involvement in CSR actions. At the same time, the involvement in actions for protecting the environment leads to a decrease in ROE, but higher ROE entails to the increase of the companies' investments towards pollution reduction activities.

Corporate governance exerts a positive influence over each variable that we have analyzed, both as a direct factor, as well as an indirect one.

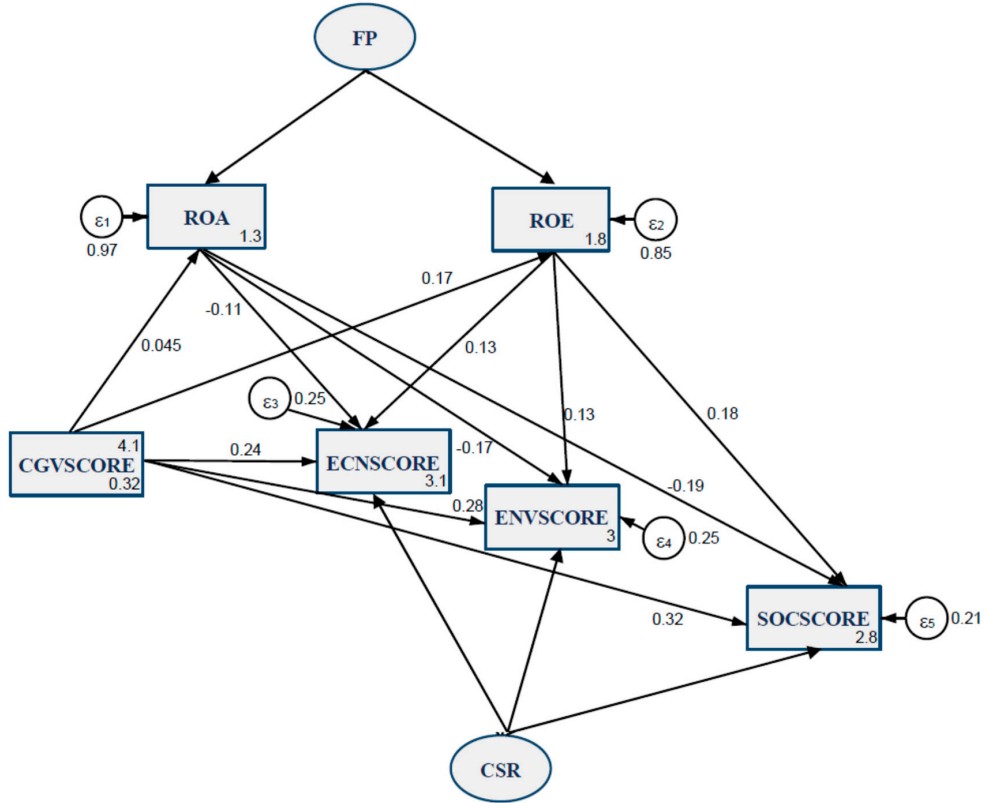

**Figure 6.** SEM results for CG-FP-CSR. Source: own research in Stata.

## 5. Concluding Remarks

This research brings a significant contribution to the existing knowledge that is currently evolving in the field of corporate social responsibility, by providing a methodological, an integrative approach, the structural equation modeling (SEM), to studying the interdependencies (direct, indirect, and total) between each dimension of CSR (economic, environmental, and social), the acknowledged performance indicators (ROA, ROE), and the corporate governance (CG).

Considering the CSR-FP binom as the central axis of our study, the results of this research have been generated within a gradual way, starting from the partial equation models, and ending with a global model comprising global interdependencies between variables. Significant dependencies, statistically validated, have led to mixed conclusions regarding the way in which the variables intercorrelate, generating both positive dependencies, as well as negative ones. The economical and financial implications of these interlinkages shed light on the significance of the decisional factors that can be found at the level of a company.

By applying the accounting measures ROA and ROE, based on the results of the general SEM model, we have identified that there are significant differences, on the one hand, in terms of the links between ROA and CSR dimensions, and, on the other hand, between ROE and CSR dimensions. As both ROA and ROE are indicators of financial performance, the results seem contradictory. Assessing these interlinkages, whether positive or negative, are involving in depth approaches, taking into account the different economic structure of the assets and equity and, mainly, the share between the assets and the equity at the company's level.

We consider that maintaining the same type of links between both ROA and ROE, and the CSR dimensions (economic, environmental, social), it can be reached only if there is a relatively constant debt ratio at the level of all the companies in the sample. However, as debt ratio greatly differs from

one company to another, the two indicators of financial performance (ROA and ROE) have different evolutions, which also influence the type of relationship with each of the CSR dimensions.

The mixed links between ROA, ROE, and each of the CSR dimensions imply—from an accounting point of view—the existence of a different degree of debt ratio from one company to another. Thus, the degree of debt ratio becomes itself an influencing factor on the financial performance.

The results of the global SEM model reconfirm the theories [11,28] that consider CSR to be a multidimensional concept, comprised of heterogenous elements that interact in different ways with other factors at the level of a company. The different natures and complexity of each CSR dimension have led to the formation of different statistical links with each of the two financial performance indicators (ROA and ROE). In this regard, our research points towards an area of mixed results [31] among CSR-FP, refuting those studies that proved a single type of link between variables, either positive [11,21], or negative [7,28]. We must also note that our research confirms corporate governance as a mediating factor of the relationship between the dimensions of CSR and financial performance indicators [9,10].

The results of this study point, as future research directions, towards extending the field of performance indicators analyzed in connection to the dimensions of CSR (stock exchange performance indicators, intellectual capital of a company, research-development and innovation etc.), since not every social performance leads to profit, and not every financial performance indicator is based on the decision to invest in CSR activities. Moreover, an in-depth approach regarding the economic structure of the assets and equity and, mainly, the share between the assets and the equity at the company's level, also by including their debt ratio influences, becomes a significant future research direction.

**Author Contributions:** All authors contributed equally to this work. I.A. and M.S (Marian Siminica). collected the data and drafted the paper; G.G.N. analyzed the data; M.C. and M.S (Mirela Sichigea) reviewed related studies. All authors wrote, reviewed, and commented on the manuscript. All authors have read and approved the final manuscript.

**Funding:** This research received no external funding.

**Conflicts of Interest:** The authors declare no conflict of interest.

## Appendix A

**Table A1.** (**a**) Wald test for SEM models, CG-CSR-FP. (**b**) Wald test for SEM models, CG-FP-CSR.

| (a) | | | |
|---|---|---|---|
| **Variables** | **Chi2** | **Df** | ***p* -Value** |
| Log_ROA | 59.93 | 4 | 0.0000 |
| Log_ROE | 54.41 | 4 | 0.0000 |
| Log_ECNSCORE | 216.13 | 1 | 0.0000 |
| Log_ENVSCORE | 280.86 | 1 | 0.0000 |
| Log_SOCSCORE | 468.77 | 1 | 0.0000 |

H0: all coefficients excluding the intercepts are 0. We can thus reject that null hypothesis for each equation.

| (b) | | | |
|---|---|---|---|
| **Variables** | **Chi2** | **Df** | ***p* -Value** |
| Log_ROA | 1.77 | 1 | 0.1835 |
| Log_ROE | 27.71 | 1 | 0.0000 |
| Log_ECNSCORE | 257.07 | 3 | 0.0000 |
| Log_ENVSCORE | 369.85 | 3 | 0.0000 |
| Log_SOCSCORE | 607.44 | 3 | 0.0000 |

H0: all coefficients excluding the intercepts are 0. We can thus reject that null hypothesis for each equation, with limited action on ROA.

Source: own research.

**Table A2.** Goodness-of-fit tests for SEM models—global model.

| (a) CG-CSR-FP | | |
|---|---|---|
| **Fit Statistic** | **Value** | **Description** |
| Likelihood ratio | | |
| chi2_ms (26) | 6623.286 | Model vs. saturated |
| *p* > chi2 | 0.000 | |
| chi2_bs (38) | 7643.061 | Baseline vs. saturated |
| *p* > chi2 | 0.000 | |
| Population error | | |
| RMSEA | 0.788 | Root mean squared error of approximation |
| 90% CI, lower bound | 0.000 | |
| upper bound | - | |
| Pclose | 0.000 | Probability RMSEA <= 0.05 |
| Information criteria | | |
| AIC | 30,527.643 | Akaike's information criterion |
| BIC | 30,651.282 | Bayesian information criterion |
| Baseline comparison | | |
| CFI | 0.132 | Comparative fit index |
| TLI | −2.254 | Tucker-Lewis index |
| Size of residuals | | |
| SRMR | 0.237 | Standardized root mean squared residual |
| CD | 0.274 | Coefficient of determination |
| (b) CG-FP-CSR | | |
| **Fit Statistic** | **Value** | **Description** |
| Likelihood ratio | | |
| chi2_ms (26) | 6474.796 | Model vs. saturated |
| *p* > chi2 | 0.000 | |
| chi2_bs (38) | 7643.061 | Baseline vs. saturated |
| *p* > chi2 | 0.000 | |
| Population error | | |
| RMSEA | 0.779 | Root mean squared error of approximation |
| 90% CI, lower bound | 0.000 | |
| upper bound | - | |
| pclose | 0.000 | Probability RMSEA <= 0.05 |
| Information criteria | | |
| AIC | 30,379.152 | Akaike's information criterion |
| BIC | 30,502.791 | Bayesian information criterion |
| Baseline comparison | | |
| CFI | 0.152 | Comparative fit index |
| TLI | −2.181 | Tucker–Lewis index |
| Size of residuals | | |
| SRMR | 0.299 | Standardized root mean squared residual |
| CD | 0.255 | Coefficient of determination |

Source: own research.

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
