# Peer review of "Well-Governed Sustainability and Financial Performance: A New Integrative Approach"

_sustainability, doi:10.3390/su11174562_

Round 1
Reviewer 1 Report
Dear authors, these are my suggestions
You sample selection its not clear. The sample is based on the European firms due the exhistence of Directive 95/2014. However, the Directive has been introduced only in 2017. On this point you must clarify your strategy.
In several line you introduce the concept of "standardization". However, large part of academics says that Directive impact on harmonization and not on standardization. This point is relevant and you must justify adequately your thesis. These are some papers:
Dumay, J., La Torre, M., & Farneti, F. (2019). Developing trust through stewardship: Implications for intellectual capital, integrated reporting, and the EU Directive 2014/95/EU. Journal of Intellectual Capital, 20(1), 11-39.
La Torre, M., Sabelfeld, S., Blomkvist, M., Tarquinio, L., & Dumay, J. (2018). Harmonising non-financial reporting regulation in Europe: Practical forces and projections for future research. Meditari Accountancy Research, 26(4), 598-621.
Venturelli, A., Caputo, F., Leopizzi, R., & Pizzi, S. (2017, June). The EU Directive on non-financial and diversity disclosure: Constraint or opportunity. In Proceedings of the EURAM Conference.
In conclusion, your paper is interesting from a methodological point of view but it needs some improement from an "accounting point of view"
Author Response
Dear Reviewer,
Thank you for your observations and for the opportunity to improve our manuscript!
We are very grateful for taking the time to analyse the paper and make very useful, encouraging and thoughtful comments and recommendations.
We have read the evaluation carefully and, based on the review reports, we performed significant revisions of our manuscript, as requested, highlighted with track changes into the manuscript (red-marked), respectively:
- regarding the first observation: Your sample selection it’s not clear. The sample is based on the European firms due the existence of Directive 95/2014. However, the Directive has been introduced only in 2017. On this point you must clarify your strategy, we have clarified our strategy as follows: “The decision to include in our analysis only companies from the European Economic Area (EEA) was based on the need to ensure the homogeneity and comparability of the data included in our research, and to offer findings for a future kindred/comparative research, after applying the European Union (EU) Directive 2014/95/EU [38], outset by 2017 (after our timespan analysis)” (lines 433-436).
-as regards the second observation: In several line you introduce the concept of "standardization". However, large part of academics says that Directive impact on harmonization and not on standardization. This point is relevant and you must justify adequately your thesis, we have replaced the concept of "standardization" with the “harmonization”, which is appropriate for our thesis (line 439). Also, we have expanded the presentation of the implications of the Directive 95/2014, finding significant points in the studies indicated as relevant for our research, and thus we have included them in the Reference list (positions 39-41) and we’ve inserted the main ideas in the Data and Sample Construction part (lines 437-450);
- as regards the final assessment: In conclusion, your paper is interesting from a methodological point of view but it needs some improvement from an "accounting point of view", thank you for your appreciation! We have extended our research implications from the accounting point of you (lines 798-814, 828-831).

Reviewer 2 Report
Dear Authors
I have had the opportunity to review the work which has given me a new insight into the interrelationships between CSR, PF and GC. As an author working in this field, I have to say that I have been surprised by the novelty of the contribution and the possible application of the Structural Equations technique to the Data Panel 2013-2017 with 614 large companies in the European Economic Area through the use of indicators published by the Thomson Reuters Database.
The following considerations should be made to the work:
1.-The introduction is too long. In it there is part of content that should be included in the theoretical framework.
2.- The last part of the introduction (lines 150 to 154) deserves a separate paragraph where the following parts are explained structurally.
3.- In the theoretical framework there is no sub-section referring to Financial Performance. It is an important part of the model and has no place. A sub-section 2.3 should be included explaining what financial performance is, the role it plays in the work, ways of measuring it (ROA, ROE) and its relationship with CSR and CG.
4.- At various points in the article, for example in lines 198 to 200, the authors speak of shareholders and stakeholders as independent actors. This must be modified since an important stakeholder of the company are shareholders. That is to say, the shareholders of the company are an important stakeholder of the company, therefore, they should not be named throughout the text as if they were independent.
5.-The Discussion and Conclusions section is a repetition of the results of the Structural Equations model. This must be rewritten, separating Discussion from Conclusions. In the conclusions the authors should contribute their own considerations of why these results have been obtained. They do not speak in any case of the possible causes of these encounters.
6.- Since the results are in accounting terms of ROA and ROE, the authors should make a clear statement as to why the results are positive or negative with each of the accounting ratios, going even deeper into the relationships found taking into account the components of the ratios: in one case Returns on Asset and in another Returns on Equity. In other words, why are there some relationships with Assets and others with Equity?
7.- In general terms, the article should go more deeply into the accounting and financial part, both in the theoretical framework and in the conclusions.
Best regards,
The reviewer.
Author Response
Dear Reviewer,
Thank you for your observations and for the opportunity to improve our manuscript!
We are very grateful for taking the time to analyse the paper and make very useful, encouraging and thoughtful comments and recommendations.
In accordance with the review report, we have firstly thank you for the appreciation of the novelty of our research and the application of the Structural Equations technique to the Data Panel 2013-2017 with 614 large companies in the European Economic Area. Further, we have made significant improvements, as follows:
- regarding the 1st consideration: The introduction is too long. In it there is part of content that should be included in the theoretical framework, we have reduced it, and moved significant parts to the second part, the conceptual framework of analysis, lines 197-199, 301-306, 318-323, and 358-385;
- regarding the 2nd observation related to Introduction - the last part (lines 150 to 154) deserves a separate paragraph where the following parts are explained structurally, we have made more clear this paragraph for presenting the structure of our paper (lines 152-156);
- regarding the 3rd consideration: In the theoretical framework there is no sub-section referring to Financial Performance. It is an important part of the model and has no place. A sub-section 2.3 should be included explaining what financial performance is, the role it plays in the work, ways of measuring it (ROA, ROE) and its relationship with CSR and CG, we have included sub-section 2.3. Financial Performance (FP), in which we have described the information requested (lines 220-246). The relationships between financial performance and CSR, respectively CG, are detailed also into 2.4.2 and 2.4.3 paragraphs;
- as regards the 4th appreciation: At various points in the article, for example in lines 198 to 200, the authors speak of shareholders and stakeholders as independent actors. This must be modified since an important stakeholder of the company are shareholders. That is to say, the shareholders of the company are an important stakeholder of the company, therefore, they should not be named throughout the text as if they were independent, we have made clear this appreciation throughout the paper, distinctively at lines 211-212;
- regarding the 5th consideration: The Discussion and Conclusions section is a repetition of the results of the Structural Equations model. This must be rewritten, separating Discussion from Conclusions. In the conclusions the authors should contribute their own considerations of why these results have been obtained. They do not speak in any case of the possible causes of these encounters, we have rewritten the Conclusion section and moved Discussions at the Results and Discussions section, lines 723-740;
- as regards the 6th appreciation: Since the results are in accounting terms of ROA and ROE, the authors should make a clear statement as to why the results are positive or negative with each of the accounting ratios, going even deeper into the relationships found taking into account the components of the ratios: in one case Returns on Asset and in another Returns on Equity. In other words, why are there some relationships with Assets and others with Equity?, we have extended our research implications from the accounting point of you at the Conclusion part (lines 798-814, 828-831).
- regarding the 7th consideration: In general terms, the article should go more deeply into the accounting and financial part, both in the theoretical framework and in the conclusions, we have made major improvements throughout the revised paper.
Thus, compared with the previous version of our manuscript, we have made major revisions to comply with Reviewers and Editor evaluations, and hopefully its present format would be suitable for publication in Sustainability.
Thank you for your consideration of this revised version of the manuscript!

Round 2
Reviewer 1 Report
Authors have made all necessary changes and the work could be published now.
Reviewer 2 Report
Dear Authors,
In general, the article has been improved according to the suggestions made.
With my best wishes,
Mercedes Rodríguez-Fernández
https://scholar.google.es/citations?user=KYsbmq8AAAAJ&hl=es